# Cell-free biosynthesis and engineering of ribosomally synthesized lanthipeptides

Wan-Qiu Liu[1,5], Xiangyang Ji[1,5], Fang Ba ⓘ [1], Yufei Zhang[1], Huiling Xu[1], Shuhui Huang[1], Xiao Zheng[1], Yifan Liu ⓘ [1,2,3] ✉, Shengjie Ling ⓘ [1,2,3] ✉, Michael C. Jewett[4] ✉ & Jian Li ⓘ [1,2,3] ✉

Ribosomally synthesized and post-translationally modified peptides (RiPPs) are a major class of natural products with diverse chemical structures and potent biological activities. A vast majority of RiPP gene clusters remain unexplored in microbial genomes, which is partially due to the lack of rapid and efficient heterologous expression systems for RiPP characterization and biosynthesis. Here, we report a unified biocatalysis (UniBioCat) system based on cell-free gene expression for rapid biosynthesis and engineering of RiPPs. We demonstrate UniBioCat by reconstituting a full biosynthetic pathway for de novo biosynthesis of salivaricin B, a lanthipeptide RiPP. Next, we delete several protease/peptidase genes from the source strain to enhance the performance of UniBioCat, which then can synthesize and screen salivaricin B variants with enhanced antimicrobial activity. Finally, we show that UniBioCat is generalizable by synthesizing and evaluating the bioactivity of ten uncharacterized lanthipeptides. We expect UniBioCat to accelerate the discovery, characterization, and synthesis of RiPPs.

Natural products are important sources of medical drugs[1]. Ribosomally synthesized and post-translationally modified peptides (RiPPs) are a prominent class of natural products, widely distributed in various organisms (e.g., bacteria, fungi, and plants)[2]. Unlike other classes of natural products such as nonribosomal peptides (NRPs) and polyketides (PKs), which are assembled by large multimodular synthetases[3], RiPPs are encoded in genomes, translated by ribosomes, and typically modified by auxiliary enzymes to form mature peptide products[2,4]. As a result, the prediction of new RiPPs from genomic data is faster and more predictable than NRPs and PKs with the aid of advanced bioinformatic tools[2,5,6]. This process can be further expedited by using automated platforms, as recently demonstrated by the Illinois Biological Foundry for Advanced Biomanufacturing (iBioFAB) platform[7].

To study RiPPs, one key step is to express their related gene clusters, including the precursor peptides and modification enzymes. Over the past several years, *Escherichia coli* has gained traction used as

a workhorse for the expression, identification, and production of RiPPs[2,8–10]. For example, the elucidation of the nisin biosynthetic pathway (e.g., enzymatic mechanism and biosynthesis) was performed by using *E. coli* as a platform[11–13]. The ongoing discovery and identification of new RiPPs are, however, still generally conducted via in vivo heterologous expression of the gene clusters in *E. coli*[7,14]. Despite some success, the rate at which new gene clusters are identified greatly outpaces the capacity to characterize and synthesize the actual RiPP products. This research bottleneck is at least in part due to the time-consuming and laborious steps of molecular cloning, cell cultivation, and protein purification (for in vitro reconstruction of enzymatic catalysis), as well as the potential cytotoxicity of RiPPs to the host cells[9,10]. In addition, biosynthesis and engineering of RiPPs in heterologous cell-based systems can be limited by insolubility and degradation of precursor peptides[15–17], insoluble expression of modification enzymes[18–21], and poor catalytic efficiency of the enzymes[22–25]. Therefore, a new

[1]School of Physical Science and Technology, ShanghaiTech University, Shanghai, China. [2]State Key Laboratory of Advanced Medical Materials and Devices, ShanghaiTech University, Shanghai, China. [3]Shanghai Clinical Research and Trial Center, Shanghai, China. [4]Department of Bioengineering, Stanford University, Stanford, CA, US. [5]These authors contributed equally: Wan-Qiu Liu, Xiangyang Ji. ✉e-mail: liuyf6@shanghaitech.edu.cn; lingshj@shanghaitech.edu.cn; mjewett@stanford.edu; lijian@shanghaitech.edu.cn

generation of expression systems for rapid synthesis, discovery, and study of RiPPs is needed to unlock new RiPPs from an ever-growing set of genomic data.

Cell-free gene expression (CFE), in vitro transcription and translation without the use of intact living cells[26,27], may offer exciting opportunities for studying RiPPs. CFE systems allow for easy manipulation of reaction conditions, improve mass transfer, and avoid cellular toxicity[28]. Moreover, they have been shown to accelerate design-build-test cycles for studying a variety of proteins and metabolic pathways[29–36].

In recent works, CFE systems have also been used to synthesize complex natural products from DNA inputs[37–40]. For instance, we showed that crude lysate-based *E. coli* CFE enabled the expression of two large nonribosomal peptide synthetases (NRPS, each molecular weight is ~300 kDa), which are capable of making the antibiotic valinomycin, a 36-membered cyclododecadepsipeptide[40]. In another example, CFE systems were employed to help guide nisin overproduction in vivo[41] and evaluate substrate tolerance of lasso peptide-forming enzymes[42]. Furthermore, over 1000 lasso peptide variants were rapidly synthesized in vitro with a success rate of 61%[42]. CFE systems can also be used to analyze the interactions between RiPP precursor recognition elements (RRE) and cognate lasso leader peptidase, leading to the elucidation of the lasso leader peptidolysis mechanism[43]. In addition, a reconstituted in vitro translation system (the PURE system[44]) has also been adopted to synthesize different types of RiPPs and their analogs (note that in most cases purified enzymes are added to the PURE system for RiPP modification), and could be combined with other strategies such as mRNA display to

investigate RiPP enzymology, promiscuity, and designing[45–52]. These above results provide the feasibility to establish CFE-based systems for rapid RiPP synthesis and characterization, yet efforts to develop CFE as an efficient and robust platform for studying and engineering RiPPs remain underdeveloped.

Here, we report a unified biocatalysis (UniBioCat) system for in vitro biosynthesis and engineering of RiPPs (Fig. 1). The foundational principle is that we can construct RiPP biosynthetic pathways totally in vitro using CFE to co-express the precursor peptides and modification enzymes to form mature RiPPs. As a model, we chose to develop UniBioCat with lanthipeptides, which are members of the rapidly expanding RiPP natural products featured with thioether cross-links called lanthionine (Lan) and methyllanthionine (MeLan)[53] and display a wide variety of bioactivities across antimicrobial and anticancer efficacy[10,54]. We demonstrate UniBioCat by reconstitution of a full biosynthetic pathway to synthesize salivaricin B, an antibacterial lanthipeptide naturally produced by an oral probiotic strain *Streptococcus salivarius* K12[55]. Using UniBioCat, we characterize the substrate promiscuity and synthesize a series of salivaricin B variants by engineering the core peptide. We finally expand UniBioCat to synthesize three other known and ten uncharacterized lanthipeptides with antimicrobial activity evaluated. Overall, UniBioCat provides a fast, streamlined, and promising approach to study RiPP natural products.

## Results

### Establishing the UniBioCat system for salivaricin B biosynthesis
To establish UniBioCat, we began with the crude lysate CFE system derived from *E. coli*, which has been well-developed with readily

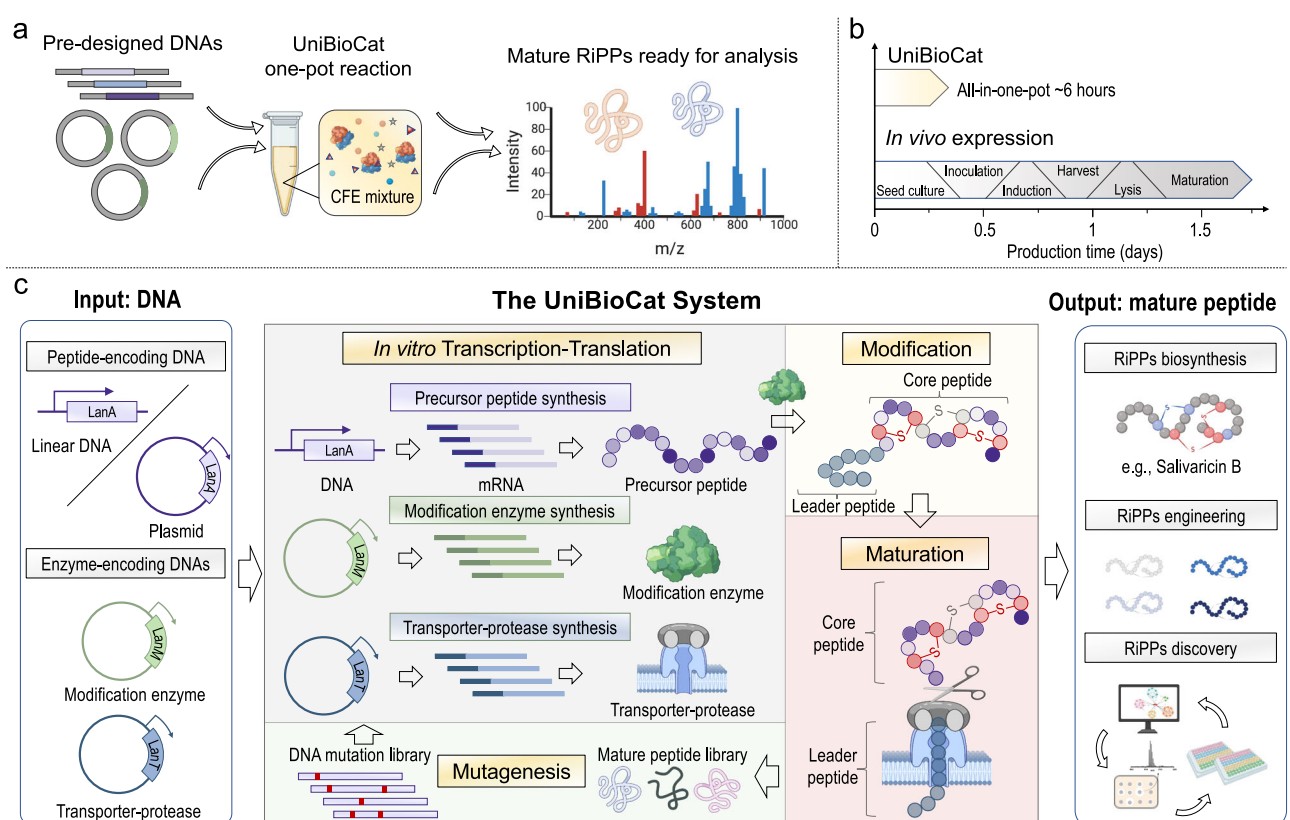

**Fig. 1 | Biosynthesis of RiPPs such as the class II lanthipeptides using the UniBioCat platform. a** Schematic diagram of the UniBioCat framework for cell-free biosynthesis of mature RiPPs from DNA as direct inputs. Created with BioRender.com. **b** Comparison of the RiPPs production timeline between UniBioCat and cell-based in vivo systems. **c** Overview of the UniBioCat workflow. By adding DNA templates of a RiPP biosynthetic pathway, UniBioCat enables in vitro transcription and translation to synthesize precursor peptide and modification enzyme(s). The enzyme(s) then modify the precursor peptide in situ for maturation, forming the final RiPP product. This integrative process can also be used to rapidly generate mutated RiPPs library. UniBioCat offers a fast and streamlined platform for RiPPs biosynthesis, engineering, and discovery. Created with BioRender.com.

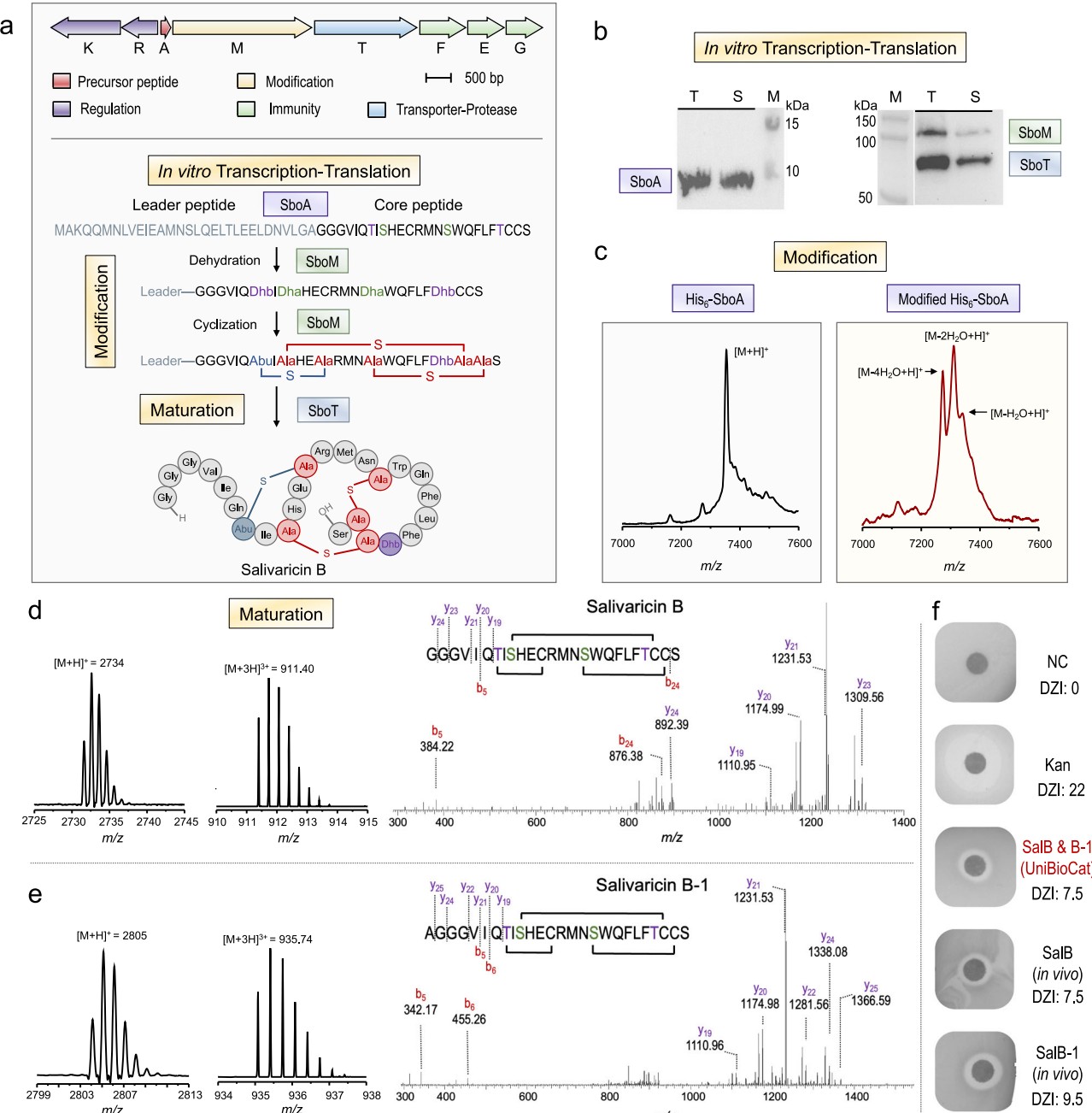

**Fig. 2 | In vitro expression, modification, and maturation of salivaricin B in UniBioCat. a** The gene cluster (top) and catalytic steps (bottom) of salivaricin B biosynthesis. Dha, dehydroalanine; Dhb, dehydrobutyrine; Abu, α-aminobutyric acid. **b** Western-blot analysis of the cell-free expressed precursor peptide (left: SboA, 7.8 kDa) and modification enzymes [right: SboM (113 kDa) and SboT (83 kDa) are co-expressed]. Peptides/enzymes are labeled with the anti-His antibody. M, protein marker; T, total protein; S, soluble protein. Results were reproduced three times independently; representative data are shown. **c** MALDI-TOF-MS analysis of the precursor peptide (left: 6x His-tagged SboA, [M + H]$^+$ $m/z$ = 7355) and modified precursor peptides (right: dehydrated 6x His-tagged SboA, [M-H$_2$O + H]$^+$ $m/z$ = 7337, [M-2H$_2$O + H]$^+$ $m/z$ = 7319, and [M−4H$_2$O + H]$^+$ $m/z$ = 7283).

**d** MALDI-TOF-MS (left), LC-MS (middle), and MS/MS (right) analysis of matured salivaricin B. **e** MALDI-TOF-MS (left), LC-MS (middle), and MS/MS (right) analysis of the analog salivaricin B−1 with one additional amino acid alanine at the $N$-terminus. **f** Antimicrobial activity assay of salivaricin B (SalB) and salivaricin B-1 (SalB-1). "SalB & B-1" was cell-free synthesized and purified for the activity assay (note that the sample used was a mixture of SalB and B-1 with a concentration of 1 mM for the test). For in vivo produced SalB and SalB-1, 1 mM of each purified SalB and SalB-1 was used for the activity assay. NC, negative control; Kan, kanamycin (50 μg/mL) as a positive control; DZI, diameter of zone of inhibition (in mm). Data shown representative of three independent experiments ($n$ = 3). Source data are provided as a Source Data file.

available standardized protocols and used by many different laboratories[56–63]. Then, we selected class II lanthipeptide salivaricin B as our model RiPP, which possesses broad antimicrobial and antiviral activities[55,64–66]. The mode of action of salivaricin B was shown to inhibit cell wall biosynthesis rather than penetrate cellular membranes[65]. In silico analysis also indicates that salivaricin B is a promising

therapeutic candidate for inhibiting SARS-CoV-2 entrance into human cells[67]. While the salivaricin B gene cluster was predicted previously[55], its biosynthetic pathway has not been fully characterized by experiments, to our knowledge. Salivaricin B biosynthesis involves three genes (Fig. 2a), including SboA (a 56-amino acid precursor peptide translated by ribosome), SboM (a bifunctional modification enzyme

that catalyzes both dehydration and cyclization of the substrate SboA), and SboT (a membrane transporter with protease activity to release the leader peptide – a process for maturation).

In vitro expression of each gene was initially performed using cell extracts prepared from *E. coli* BL21 Star (DE3). However, cell-free expressed SboA was not visible by the Western-blot analysis, and the soluble fractions of SboM and SboT were not satisfactory (Supplementary Fig. 1). Then, we switched to using chaperones-enriched cell extracts, containing DanK-DnaJ-GrpE and GroES-GroEL that might help facilitate protein folding and enhance protein solubility as described in our previous work[31,32]. By doing this, both expression levels and solubility of the three proteins were notably increased (Fig. 2b and see Supplementary Fig. 1 for the individual expression of the three genes). Of note, the improved solubility of the membrane transporter SboT is also likely due to the native membrane vesicles present in the cell extract[68]. Next, we co-expressed SboA and SboM to see if the precursor peptide SboA can be modified by SboM. Fully dehydrated SboA ($-4H_2O$) was detected by matrix-assisted laser desorption/ionization time-of-flight mass spectrometry (MALDI-TOF-MS), albeit partially dehydrated SboA ($-1$ and $-2H_2O$) was observed as well (Fig. 2c). Then, we expressed the three genes together in a single-pot reaction, after which the product was extracted for MALDI-TOF-MS and liquid chromatography-tandem mass spectrometry (LC-MS/MS) analysis. The results demonstrated a successful biosynthesis of matured salivaricin B from in vitro co-expression of the three genes (Fig. 2d). Interestingly, we also observed a salivaricin B-like peptide with one additional amino acid alanine at the *N*-terminus, which was named salivaricin B-1 (Fig. 2e). This observation is in agreement with previous studies on the LctT protease, which is responsible for the maturation of lacticin 481. But LctT also cleaves the position between glycine ($-2$) and alanine ($-1$), generating the lacticin 481 analogs with an *N*-terminal alanine[69]. However, salivaricin B-1 is not produced by *S. salivarius* K12, the natural producer of salivaricin B[55]. To further verify the cell-free synthesized salivaricin B and B-1, we synthesized both products by expressing their precursor peptides (each with a trypsin cleavage site) in *E. coli*, followed by in vitro trypsin digestion, and HPLC purification of the matured peptides (see the product preparation in "**Methods**"). The matured peptides were then analyzed by MALDI-TOF-MS and LC-MS/MS. The results showed that the molecular mass and fragmentation of in vivo produced salivaricin B and B-1 were consistent with cell-free synthesized peptides (Supplementary Fig. 2). In addition, the desired formation of three thioether rings was confirmed by the *N*-ethylmaleimide (NEM) cysteine alkylation assay[20] (Supplementary Fig. 3). Moreover, we tested the antimicrobial activity of the in vivo and cell-free produced salivaricin B and B-1 against *Staphylococcus aureus* RN4220 using the agar diffusion assay. We observed that both peptides showed clear inhibitory effects on cell growth (Fig. 2f). Taken together, these results demonstrate the synthetic capability of UniBioCat: (i) multiple proteins can be co-expressed in soluble form, ranging from a short precursor peptide (SboA, 56 amino acids) to a large membrane transporter (SboT); (ii) cell-free expressed enzymes are active to modify and mature salivaricin B; and (iii) the final peptide can be obtained in hours from the reaction for functional characterization such as antimicrobial activity assay.

### Enhancing the performance of UniBioCat by deleting protease/peptidase genes in E. coli

While we have demonstrated the concept of using UniBioCat to synthesize salivaricin B, we also observed multiple salivaricin B-like analogs in addition to salivaricin B-1 (Supplementary Fig. 4a). This could be as a result of the non-specific removal of the leader peptide (from $-7L$ to $+4V$) digested by endogenous proteases and/or peptidases from *E. coli* cell extracts. Our results were confirmed with the in vivo produced salivaricin B samples (Supplementary Fig. 4a). Adding a protease

inhibitor cocktail to UniBioCat reactions could reduce the degree of non-specific digestions (Supplementary Fig. 4b, c). Hence, this motivated us to delete potential protease/peptidase genes from the genome of the source strain *E. coli* BL21 Star (DE3) that was used to make the cell-free extracts (Supplementary Fig. 5a). To do this, we selected six endogenous protease/peptidase genes (Supplementary Table 4), which were individually deleted by λ-Red mediated recombination[70]. Using each engineered strain, we prepared new cell extracts for in vitro salivaricin B biosynthesis (note that all cell extracts were additionally enriched with the modification enzyme SboM to simplify the UniBioCat system and focus on SboA translation). After UniBioCat reactions, we observed that, in general, deletion of each individual protease/peptidase gene was beneficial for reducing non-specific degradation of the leader peptide, although the degradation was not completely inhibited (Supplementary Figs. 5b, c, and 6). We next combined deletion of two genes (*degP* and *pepN* encode a protease and a peptidase, respectively) to generate source strain that performed better in CFE reactions than lysates with a single deletion of *degP* or *pepN* alone (Supplementary Fig. 5c). This double-deletion strain was used to prepare cell extracts for UniBioCat reactions in all remaining experiments. We anticipate that protease-deleted strains might be used as generic chassis for CFE to synthesize many other RiPPs (and even other classes of short peptides).

### Generating salivaricin B variants by site-specific mutation of the core peptide

Next, we aimed to rapidly generate salivaricin B variants by using the UniBioCat system. Previous studies showed that some class II lanthipeptide synthetases are tolerant to substrate promiscuity[71–73]. This property enables the engineering of lanthipeptide core sequences to generate new/close-to-nature lanthipeptides with potentially improved or new biological activities, such as antimicrobial activity[74]. As an example, the tolerance of the biosynthetic machinery from the so-called lacticin 481 families (belonging to the class II lanthipeptides) has been extensively demonstrated to generate a large pool of variants[75–78]. Similarly, using the same modification enzyme (NukM), nukacin ISK-1 (a 27-residue peptide) could be engineered to yield hundreds of variants, covering every position in the peptide, with two mutants of increased antimicrobial potency[76]. The lacticin 481 biosynthetic machinery (LctM) was even able to modify a protein-fused precursor peptide (Aga2-LctA), allowing for yeast surface display to screen variants with new activities[78]. However, substrate tolerance of the modification enzyme SboM has not yet been investigated in salivaricin B biosynthesis.

To evaluate the catalytic promiscuity of SboM, we initially engineered the core peptide sequence of salivaricin B by site-specific mutation. We first performed a BLAST-P analysis using the precursor peptide of salivaricin B as a query sequence to define the position(s) for mutagenesis. We identified two positions (Q6 and L20) that are not highly conserved on the core peptide (Supplementary Fig. 7). Then, we carried out site-directed saturation mutagenesis of the two positions, respectively, leading to a library of 38 gene expression templates. Using UniBioCat, 34 out of 38 variant precursor peptides were successfully expressed (see Western-blot in Supplementary Fig. 8). After purification, we ran MALDI-TOF-MS analysis and observed that 31 variants were detectable from the 34 purified samples (Supplementary Fig. 9). We then co-expressed SboT with each of the 31 variant precursor peptides in the SboM-enriched UniBioCat system and then analyzed each matured salivaricin B variant by MALDI-TOF-MS. The results indicated that 31 matured variants were detected with correct molecular masses, which are consistent with the calculated masses (Fig. 3a and Supplementary Fig. 10). This observation suggests that SboM is active to catalyze the dehydration and cyclization of mutated precursor peptides (substrates) with at least a single-site mutation (Q6 or L20).

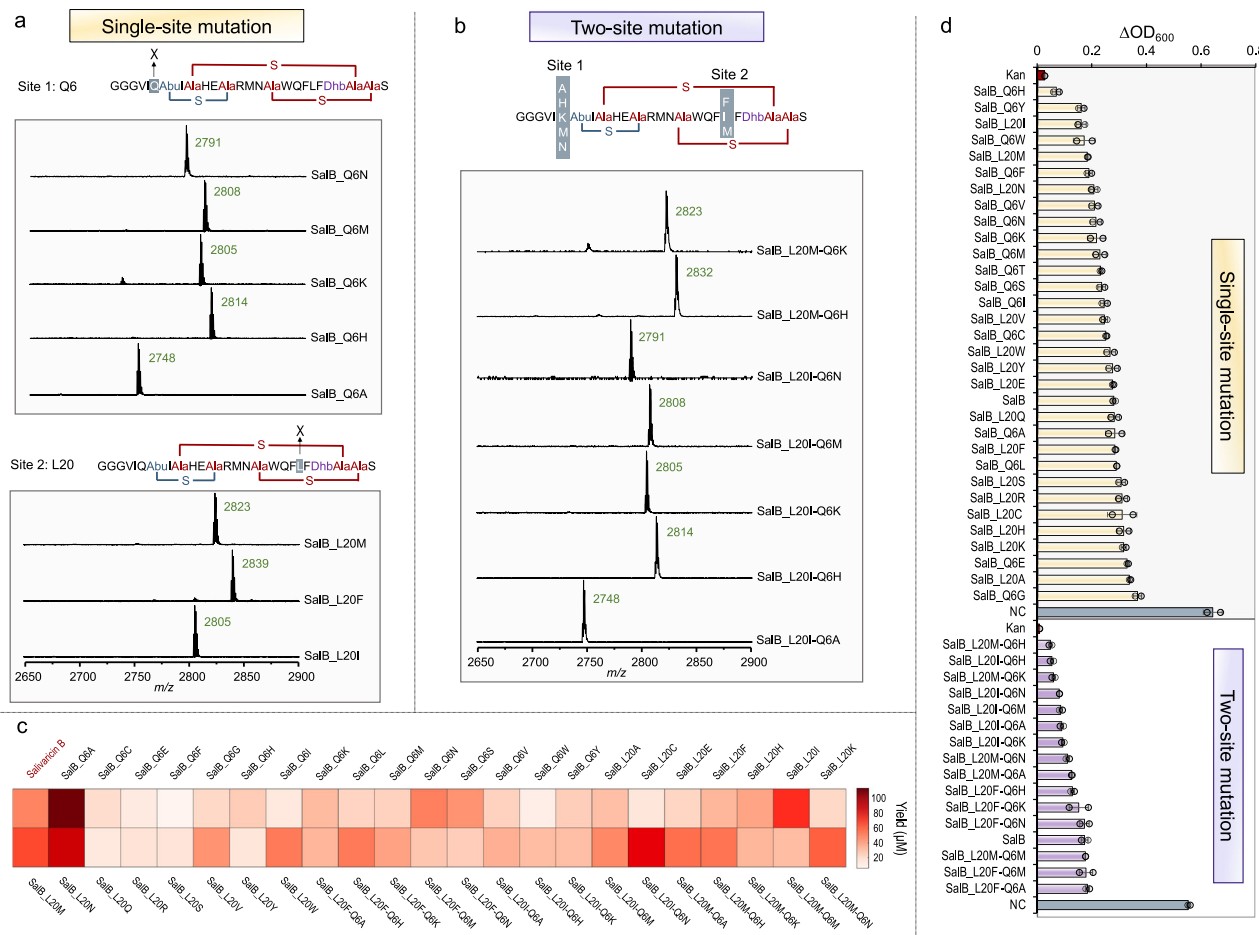

**Fig. 3 | Generation of salivaricin B mutants and their antimicrobial activity.** **a** Representative MALDI-TOF-MS analyses of single-site mutagenesis at the position of Q6 or L20. **b** Representative MALDI-TOF-MS analyses of two-site mutagenesis at the positions of Q6 and L20. In (**a**) and (**b**), salivaricin B-1 based variants are the most abundant products in all analyses, which are indicated with the [M + H]⁺ ions. **c** Quantification of salivaricin B mutants that are synthesized in UniBioCat reactions (see Supplementary Table 5 for the yield of each peptide variant). **d** Antimicrobial

activity assay showing $\Delta OD_{600}$ profiles of *S. aureus* RN4220 growth treated with single-site (top) and two-site (bottom) mutants. Note that cell-free reaction mixture is used directly for the assay without purification of each mutant. NC, negative control without gene templates in the cell-free reaction; Kan, kanamycin (25 μg/mL) as a positive control. Data shown representative of three independent experiments ($n = 3$). In (**d**), data are presented as mean±s.d. of two independent experiments ($n = 2$). Source data are provided as a Source Data file.

We next sought to test the antimicrobial activity of the mutated salivaricin B variants against the indicator strain *S. aureus* RN4220. To compare their activity, we first determined the concentrations of all peptide variants synthesized in UniBioCat reactions (Fig. 3c and Supplementary Table 5) and added an equal amount of each variant to the cell culture in 96-well plates. During the cultivation, $OD_{600}$ values were measured every 2 h for a total of 12 h. The data showed that 19 out of 31 variants exhibited stronger inhibition of cell growth than salivaricin B (Fig. 3d, top panel). Based on the single-site mutagenesis and activity assay, we performed a second-round mutation with two-site mutagenesis at both Q6 and L20 using eight selected variants (the combination leads to 15 two-site mutated variants). All these new variants could be synthesized and detected by MALDI-TOF-MS (see Fig. 3b for representative analyses). By screening their antimicrobial activity, we found that 11 out of the 15 two-site mutants displayed stronger inhibition, while the remaining four variants showed a similar level of inhibitory effect on cell growth as compared to salivaricin B (Fig. 3d, bottom panel). To further verify their activity and compare to standard in vivo approaches, six variants including single- and two-site mutation were selected to be produced by in vivo expression/modification and in vitro digestion for maturation. These matured products were purified and confirmed by MALDI-TOF-MS and LC-MS/MS (Supplementary Fig. 11). Then, we tested their bioactivity and observed that they all

showed notably stronger inhibition than salivaricin B within 10 h of cultivation (Supplementary Fig. 12). Minimal inhibitory concentration (MIC) and $IC_{50}$ assay were also performed and the results suggested that $IC_{50}$ values of salivaricin B and the variants were more than 8 μM, which is in line with a previous report ($IC_{50} = 8.64$ μM)[65]. Additionally, we evaluated the MIC and $IC_{50}$ of four strong variants (see Supplementary Fig. 12 for SalB_Q6H, SalB_L20M, SalB_L20I-Q6H, and SalB_L20M-Q6H) against three other indicator strains, including *Bacillus subtilis*, *Micrococcus luteus*, and *Lactococcus lactis*. The data indicated that the four variants generally showed an increased inhibitory effect on the indicators (Supplementary Table 6). Notably, the strongest variant SalB_Q6H also performed the best against the three indicators with MICs and $IC_{50}$ values reduced by 75% and 65-72%, respectively, as compared to the wild-type salivaricin B.

Taken together, the above results demonstrate that the UniBioCat system is a feasible and efficient platform to characterize the catalytic promiscuity of RiPP modification enzymes (here is SboM without purification), synthesize multiple RiPP variants, and screen the ones with enhanced bioactivity. Furthermore, we found that although salivaricin B variants could be detected, salivaricin B-1 based variants were the most abundant products in all analyses (Fig. 3a, b), which might be due to the non-specific cleavage of SboT. To test this hypothesis, we mutated the cleavage site of SboT at the positions of G-2 or A-1

(Supplementary Fig. 13a). The results showed that SboT has relatively relaxed activity toward the cleavage site (GAG) and is more likely to cleave between the residues of G-2 and A-1 in UniBioCat reactions, leading to more peptide products with an additional amino acid (A) at the *N*-terminus (Supplementary Figs. 13b and 14). However, such "A-plus peptide (salivaricin B-1)" showed an even higher antimicrobial activity than salivaricin B (Fig. 2f and Supplementary Fig. 12). Yet, additional studies are necessary to further characterize the catalytic function and substate tolerance of SboT in the future.

### Expanding UniBioCat to discover and biosynthesize uncharacterized lanthipeptides

To showcase the generalizability of the UniBioCat approach, we next sought to search for uncharacterized lanthipeptides from the genome database and synthesize them in our cell-free system. To achieve this, we designed the following strategies. First, given the broad substrate tolerance of SboM, we decided to use the leader peptide of SboA to guide SboM in modifying the selected core peptides. That is, we fused the leader peptide to the *N*-terminus of different other core peptides to create hybrid precursor peptides, which are likely modified by SboM. This strategy for generating natural and/or hybrid RiPP variants has been well documented in previous studies[79,80]. Second, SboT was co-expressed in UniBioCat reactions to mature lanthipeptides by removing the leader peptide. Third, to simplify the process of gene template construction, we generated linear DNA templates by overlap PCR that basically contain a T7 promoter, a gene sequence of each hybrid precursor peptide, and a terminator, which can be used directly for cell-free gene expression. Note that extra non-coding DNA sequences (~300 bp from the pJL1 plasmid backbone) upstream of the promoter and downstream of the terminator were added to each linear template, respectively, to reduce the potential degradation of DNA sequences by exonucleases in the CFE reactions. Finally, we mined the NCBI genome database (using the BLAST-P tool) to search for potential candidate peptides. We used the amino acid sequence of the precursor peptide of salivaricin B as a query to search against the non-redundant protein sequences database (NCBI). Then, we selected twelve candidates of lanthipeptides (over 45% amino acid identity of precursor peptides) from different microbial genomes (e.g., *Actinomyces*, *Bacillus*, *Bifidobacterium*, *Clostridium*, *Gardnerella*, and *Leuconostoc*, etc.) for cell-free biosynthesis (named Lan 1 - Lan 12; Fig. 4a). For comparison, we also constructed five linear gene templates by using the known class II lanthipeptides, including lacticin 481[81], ruminococcin A[82], nukacin ISK-1[83], mutacin II[84], and salivaricin A[85].

After UniBioCat reactions, all cell-free synthesized lanthipeptides were analyzed by MALDI-TOF-MS and LC-MS/MS. We observed that 10 out of the 12 candidate lanthipeptides (with 65-85% amino acid identity to the core peptide of salivaricin B) were detected with correct molecular masses and fragmentation patterns (Fig. 4a and Supplementary Fig. 15). For the five known lanthipeptides, three of them (i.e., lacticin 481, ruminococcin A, and nukacin ISK-1) were also synthesized and detected (note that no fragmentation was observed for nukacin ISK-1 in LC-MS/MS, perhaps due to low abundance). Salivaricin A was not synthesized using the salivaricin B-based modification system. This is probably because salivaricin A has its own modifying machineries independent of salivaricin B (salivaricin A and B are simultaneously produced in the same native producer)[55,85]. Next, we evaluated the antimicrobial activity of the newly synthesized lanthipeptides. We found that they all showed notable inhibition on the growth of *S. aureus* RN4220 as compared to the negative control without gene templates, albeit their inhibitory levels varied (Fig. 4b and Supplementary Fig. 16). These lanthipeptides displaying antimicrobial activity is probably due to the fact that they all belong to the same structural family of the lantibiotic lacticin 481[75,77]. Taken together, our results suggest that UniBioCat is capable of synthesizing different lanthipeptides via rational design. Moreover, we found that cell-free synthesized

products without purification as a crude reaction mixture can be used directly in the activity assay. This saves time and cost to help expedite the process of discovering and identifying bioactive lanthipeptides.

## Discussion

We demonstrated a CFE-based UniBioCat system that integrates transcription-translation, modification, and maturation (plus mutagenesis) for rapid biosynthesis and engineering of RiPP natural products. As a model, we established the UniBioCat platform by reconstituting the biosynthetic pathway to synthesize salivaricin B. Then, we showed the platform's ability to synthesize a library of 46 matured salivaricin B variants from a set of 53 designs (i.e., 38 single- and 15 double-site mutations). To demonstrate the general nature of the UniBioCat approach, we further constructed hybrid peptides by fusing the leader peptide of SboA to the core peptide of uncharacterized RiPPs (e.g., class II lanthipeptides), demonstrating that 10 out of 12 candidate RiPPs (ca. 83%) were successfully synthesized using the UniBioCat system. Although cell-free synthesized RiPPs could be purified for their activity assay, this purification step was not necessary for functional testing because the antimicrobial activity could be readily monitored from the crude reaction mixtures. However, one should also note that if partially modified (precursor) peptides are present in the mixture and their bioactivity cannot be ruled out, purification of the fully modified, matured peptides might be required to further verify their activity.

The UniBioCat platform has several key features. First, it unifies the entire process for de novo biosynthesis of complex RiPP natural products, making it easy to use since the four main steps (transcription, translation, modification, and maturation) can be completed in a single pot just by adding genes (plasmids or linear DNA templates). Second, our cell-free approach is fast. It requires only hours to obtain matured RiPPs; however, several days or weeks may otherwise be needed to cultivate native strains or heterologous hosts for conventional product production approaches. Using a crude reaction mixture for bioactivity assay also reduces the production time by eliminating the RiPP purification stage. Third, the UniBioCat system is robust. Our results indicate that multiple proteins (the precursor peptide and modification enzymes) can be simultaneously expressed; enzymes are active to catalyze post-translational modifications (i.e., dehydration and cyclization) by SboM and proteolysis by SboT; and the final matured RiPPs are formed with correct structures and molecular weights. In addition to the above features, UniBioCat might also be constructed by taking advantage of other CFE systems[28,86], which can better mimic the cellular endogenous environment of native hosts for efficient gene expression such as proper codon usage and correct protein folding.

Over the past decade, the study of RiPPs has been significantly advanced with powerful prediction tools, high-throughput platforms, and a more thorough understanding of the biosynthetic mechanisms underlying RiPP formation[2,4,5]. In this context, in vivo heterologous expression systems play a core role in the research process, yet often encounter cellular toxicity and expression issues (e.g., insolubility and instability)[15–25]. Recently, cell-free systems have been raised to address the challenges to a large extent. Without using living cells, CFE is naturally suitable for synthesizing toxic compounds including RiPPs[37]. Previous studies have shown that CFE reaction conditions can be readily optimized and controlled due to their open environment, resulting in high-quality and soluble expression of proteins/ enzymes[26,27,30–32]. Here we also tuned the UniBioCat system by just using chaperones-enriched cell extracts to increase the solubility of both precursor peptide (SboA) and modification enzymes (SboM and SboT) (Supplementary Fig. 1). Note that no stabilization tags (e.g., SUMO and MBP) were fused to the above three peptide/proteins for solubilization. However, these tags are normally needed for soluble expression of the RiPP biosynthetic pathway in vivo[9,10,16,20].

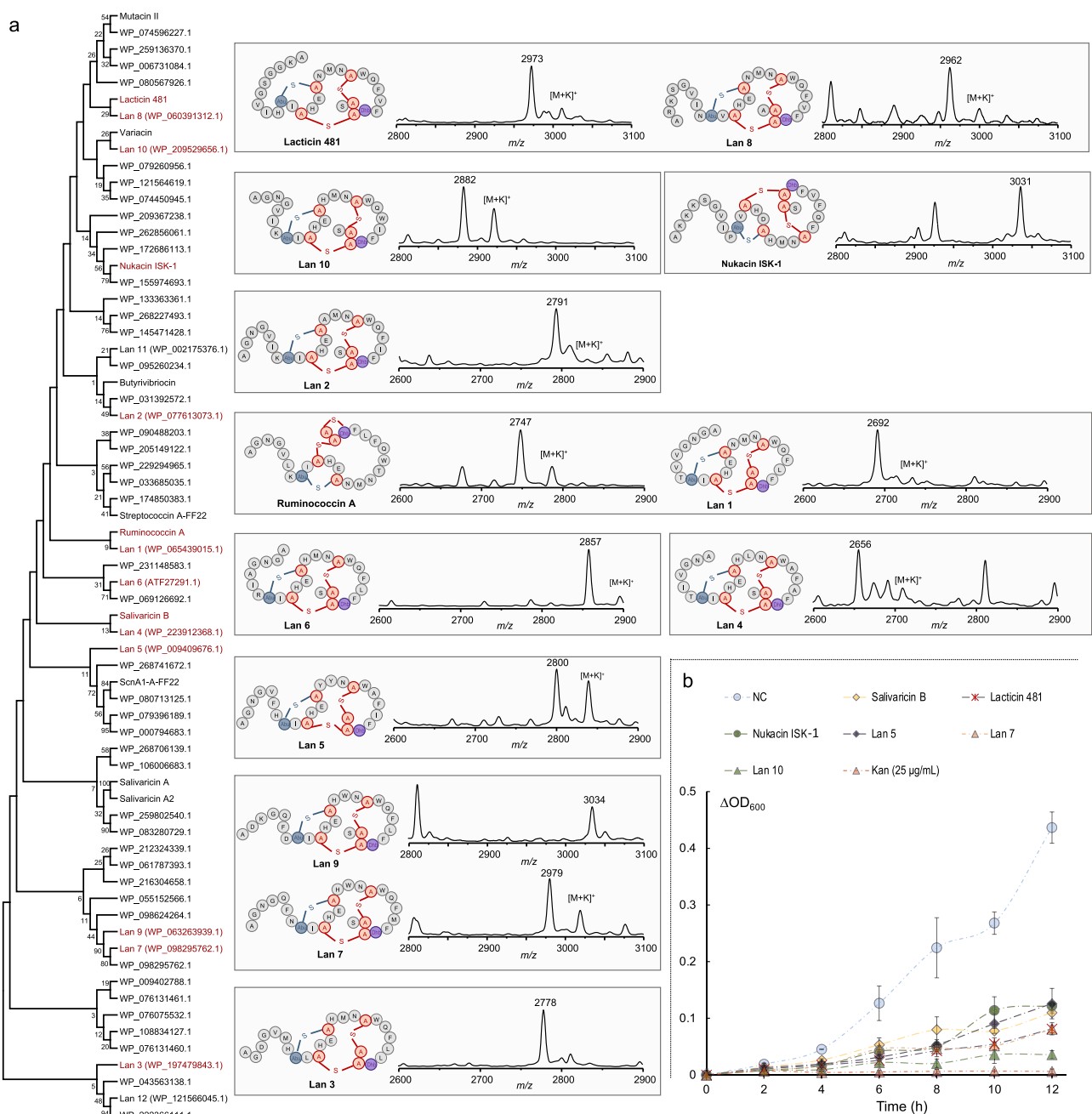

**Fig. 4 | UniBioCat enables discovery and biosynthesis of uncharacterized lanthipeptides. a** Genome mining, cell-free biosynthesis, and MALDI-TOF-MS identification of uncharacterized lanthipeptides. Amino acid sequences of Lan 1 - Lan 12 are obtained from NCBI under the accession numbers shown in the brackets. The leader peptide of SboA is fused to the *N*-terminus of each selected core peptide, allowing the two biosynthetic enzymes (SboM and SboT) to produce mature lanthipeptides (e.g., Lan 1 - Lan 10). Note that in (**a**) all lanthipeptide sequences are presented with one additional amino acid alanine at the *N*-terminus of the final peptide, which is indicated with the [M + H]⁺ ions. **b** Antimicrobial activity assay

showing $\Delta OD_{600}$ profiles of *S. aureus* RN4220 growth treated with cell-free synthesized products including three representative new lanthipeptides (i.e., Lan 5, Lan 7, and Lan 10; see Supplementary Fig. 16 for the activity assay of all tested lanthipeptides). Note that cell-free reaction mixture is used directly for the assay without purification of each lanthipeptide. NC, negative control without gene templates in the cell-free reaction; Kan, kanamycin (25 μg/mL) as a positive control. Data shown representative of three independent experiments (*n* = 3). In (**b**), data are presented as mean±s.d. of two independent experiments (*n* = 2). Source data are provided as a Source Data file.

Furthermore, non-specific degradation of the leader peptide could be reduced, albeit not completely blocked, by deleting potential protease genes of the source strain (Supplementary Fig. 5). This suggests that the construction of protease-deficient strains in future work will further help establish a better UniBioCat platform for the efficient biosynthesis of RiPPs.

Like the promiscuous property of some class II lanthipeptide synthetases (LanM)[71–73], we found that the modification enzyme

(SboM) is also tolerant to substrate promiscuity, enabling the successful formation of nearly 50 salivaricin B variants. Interestingly, several variants showed a stronger antimicrobial activity than the wild-type salivaricin B (Fig. 3d). In particular, the strongest variant performed the best against the indicator strains with notable reduction of the MICs and $IC_{50}$ values by 75% and 65-72%, respectively (Supplementary Table 6). While we did not attempt to generate thousands of variants or more using UniBioCat, our platform is capable of achieving

this scalable biosynthesis in principle, which has been demonstrated in previous studies with other cell-free systems[41,42]. Our results suggest that UniBioCat not only enables characterization of SboM's catalytic promiscuity and thus generation of salivaricin B variants but also allows screening of the ones with enhanced antimicrobial activity. Moreover, we were able to use UniBioCat to synthesize ten uncharacterized lanthipeptides homologous to salivaricin B and test their bioactivity without purification (Fig. 4). This highlights the advantage of CFE systems for discovering bioactive RiPPs because such method can rapidly distinguish whether RiPPs are antimicrobial or not, allowing the selection of bioactive peptides for further detailed study if necessary.

In this work, the class II lanthipeptide salivaricin B was chosen as one example to demonstrate UniBioCat with the success of de novo biosynthesis, engineering, and screening. While many other classes of RiPPs requiring various modifications are not investigated in the present study, we believe that UniBioCat can be expanded to more RiPPs in the future as a few RiPPs like thiopeptides and lasso peptides have been synthesized by different CFE systems (e.g., the PURE system) as well[42,48].

From the point of view of production, one should note that UniBioCat is currently performed on a microliter scale, which is not sufficient for product purification from just one standard reaction (normally 15 µL). This limitation might be overcome by further scaling up the reaction volume or using a semi-continuous reaction format[26,87,88]. Nonetheless, the small-scale reaction mixture has been shown to be sufficient for bioactivity screening (Figs. 3d and 4b). Similar to our work, a recent study also demonstrated the feasibility of combining CFE and deep learning for de novo design, rapid production, and screening of bioactive antimicrobial peptides (AMPs), which are also performed in microliter scales (note that these AMPs do not require post-translational modifications)[89].

Looking forward, we anticipate that UniBioCat together with other high-throughput platforms such as automated workstations will help facilitate the study of RiPP natural products. UniBioCat holds promise to play an important role in the study, engineering, and biosynthesis of RiPP compounds to meet the increasing demand for bioactive natural product discovery and development.

## Methods

### Bacterial strains and media
*E. coli* DH5α was used for molecular cloning and plasmid propagation. *E. coli* BL21 Star (DE3) and derived strains were used for in vivo expression and cell extract preparation. LB (Luria-Bertani) medium (10 g/L tryptone, 5 g/L yeast extract, and 10 g/L sodium chloride) was used for *E. coli* cultivation. TB (Terrific Broth) medium [12 g/L tryptone, 24 g/L yeast extract, 0.4% glycerol (v/v), 2.31 g/L potassium dihydrogen phosphate, and 12.54 g/L potassium hydrogen phosphate] was used for in vivo protein expression. 2xYTPG medium (10 g/L yeast extract, 16 g/L tryptone, 5 g/L sodium chloride, 7 g/L potassium hydrogen phosphate, 3 g/L potassium dihydrogen phosphate, and 18 g/L glucose, pH 7.2) was used to grow cells for cell extract preparation.

### Construction of expression templates
All genes were codon-optimized and chemically synthesized by GENEWIZ (Suzhou, China). The gene sequences are listed in Supplementary Table 1. For in vivo expression, a trypsin-digested site (R) was inserted at *N*-terminus of the core peptide, allowing subsequent in vitro maturation by trypsin digestion. The plasmid pRSFDuet-1 was used for in vivo co-expression of SboA and SboM by inserting the *sboA* gene between *BamH*I and *Hind*III (located in the MCSI site) and the *sboM* gene between *Nde*I and *Xho*I (located in the MCSII site), respectively. For CFE expression, the gene of *sboA* was cloned into the plasmid of pJL1 (Addgene #69496) and *sboM* and *sboT* were cloned into pET28a, respectively. To generate salivaricin B variants, the sites at Q6 and/or

L20 of the core peptide were mutated by PCR amplification with mutated primers. Then, each mutated sequence was ligated with the pJL1 backbone using homologous recombination. To synthesize uncharacterized RiPPs (i.e., Lan 1 - Lan 12), linear DNA templates were used for CFE reactions. Each linear template was constructed by overlap PCR ligation of a T7 promoter (flanked with a ~300 bp upstream sequence amplified from the pJL1 backbone), the leader peptide sequence of SboA, a core peptide sequence of Lan 1 - Lan 12 (generated directly with two primers), and a T7 terminator (flanked with a ~300 bp downstream sequence amplified from the pJL1 backbone). All primers and plasmids used in this study are listed in Supplementary Table 2 and Table 3, respectively.

### In vivo expression and preparation of standard mature peptides
The plasmids pRSFDuet-1-*sboA*_A-1R and pRSFDuet-1-*sboA*_G-2R were used for precursor expression. The plasmids pRSFDuet-1-*sboA*_A-1R-*sboM* and pRSFDuet-1-*sboA*_G-2R-*sboM* were used for expressing modified precursors of salivaricin B and salivaricin B-1, respectively. *E. coli* BL21 Star (DE3) harboring each corresponding plasmid was used for the production. Cells were initially cultivated in LB overnight and then 15 mL of the overnight preculture was used to inoculate 1 L TB medium. When the $OD_{600}$ reached 0.6-0.8, cells were induced with 0.5 mM isopropyl-β-D-1-thiogalacopyranoside (IPTG), followed by cultivation at 37 °C for 3 h (for precursor production) or at 22 °C for 20 h (for modified precursor production). Purification of precursor peptides was performed as reported previously with slight modifications[20,90]. After cultivation, cells were harvested by centrifugation at 5000 g for 10 min and washed with start buffer (20 mM $Na_2HPO_4$, 500 mM NaCl, 0.5 mM imidazole, and 10% glycerol, pH 7.5). The pelleted cells were resuspended in denaturing buffer LanA I (20 mM $Na_2HPO_4$, 500 mM NaCl, 0.5 mM imidazole, 6 M guanidine hydrochloride, and 10% glycerol, pH 7.5) and lysed by sonication (60% amplitude, 10 s on/off for a total of 30 min). The lysate was then centrifuged at $20,000 \times g$ and 4 °C for 1 h. After centrifugation, the supernatant was filtered with 0.22 µM filter membrane and then loaded to a HiTrap HP nickel affinity column. The column was first washed with denaturing buffer LanA II (20 mM $Na_2HPO_4$, 500 mM NaCl, 30 mM imidazole, 4 M guanidine hydrochloride, and 10% glycerol, pH 7.5) and then eluted with elution buffer (20 mM $Na_2HPO_4$, 500 mM NaCl, 500 mM imidazole, 4 M guanidine hydrochloride, and 10% glycerol, pH 7.5). The eluted samples were desalted by dialysis (3.5 kDa molecular weight cut off, MWCO) with desalination buffer (20 mM $Na_2HPO_4$ and 25 mM NaCl, pH 7.5). After desalination, the peptides precipitated from the buffer and then were collected by centrifugation for the following analysis and maturation. Desalted samples were analyzed by matrix-assisted laser desorption/ionization time-of-flight mass spectrometry (MALDI-TOF-MS). Samples of modified precursors were further digested by trypsin (37 °C, 2 h) to remove the leader peptide, generating matured peptides.

Precursors (unmodified or modified peptides) and matured peptides were further purified by reverse phase HPLC (300SB-C3 Semi-Prep HPLC Column, 9.4 mm × 250 mm). Samples were separated by solvent A (water/0.1% TFA) and solvent B (100% ACN/0.1% TFA). Solvent B was set at 10% for 5 min, ramped up to 70% in 50 min, and increased to 100% for 2 min at a flow rate of 2 mL/min. Fractions corresponding to UV absorbance peaks (210 nm) were collected and identified by MALDI-TOF-MS. All purified products were evaporated, lyophilized, and stored at −80 °C until use. The resultant salivaricin B and salivaricin B-1 are used as standard for comparison.

### Protease/peptidase deletion
Six endogenous proteases and peptidases deleted from *E. coli* BL21 Star (DE3) are listed in Supplementary Table 4. The protease/peptidase genes were deleted by λ-Red mediated recombination as described previously with slight modifications[70]. Briefly, linear deletion cassettes

consisting of "homologous fragment (left)-attP-resistance gene (kan)-attB-homologous fragment (right)" were constructed from the plasmid pKD4-TP901-1-attP/attB. Temperature-sensitive plasmid pKD46 was firstly transformed into *E. coli* BL21 Star (DE3) for the expression of λ-Red recombinases. Then, each linear deletion cassette was individually transformed into *E. coli* BL21 Star (DE3) containing expressed λ-Red recombinases for homologous recombination. Finally, the plasmid pYF2 for expression of the integrase TP901-1 was transformed into the above strain for resistance elimination. Deletion of the target genes was verified by PCR and sequencing.

### Cell extract preparation

Cell growth, collection, and extracts were prepared as described previously[40]. Briefly, all *E. coli* strains were grown in 2xYTPG medium. In each cultivation, 1 L of 2xYTPG was inoculated with overnight pre-culture at an initial $OD_{600}$ of 0.05. When the $OD_{600}$ reached 0.6-0.8, cells were induced with 1 mM IPTG to express T7 RNA polymerase, followed by harvest at an $OD_{600}$ of 3.0. Then, cells were washed three times with cold S30 Buffer (10 mM Tris-acetate, 14 mM magnesium acetate, and 60 mM potassium acetate). After the final wash and centrifugation, the pelleted cells were resuspended in S30 Buffer (1 mL/g of wet cell mass) and lysed by sonication (10 s on/off, 50% of amplitude, input energy ~600 Joules). The lysate was then centrifuged twice at $12,000 \times g$ and 4 °C for 10 min. The resulting supernatant was flash frozen in liquid nitrogen and stored at −80 °C until use.

### Cell-free gene expression (CFE) reactions

CFE reactions were performed to express peptides/enzymes in 1.5 mL microcentrifuge tubes. A standard reaction (15 μL) contained the following components: 12 mM magnesium glutamate, 10 mM ammonium glutamate, 130 mM potassium glutamate, 1.2 mM ATP, 0.85 mM each of GTP, UTP, and CTP, 34 μg/mL folinic acid, 170 μg/mL of *E. coli* tRNA mixture, 2 mM each of 20 standard amino acids, 0.33 mM nicotinamide adenine dinucleotide (NAD), 0.27 mM coenzyme A (CoA), 1.5 mM spermidine, 1 mM putrescine, 4 mM sodium oxalate, 33 mM phosphoenolpyruvate (PEP), 5-15 nM gene template (plasmid or linear DNA), and 27% (v/v) of cell extract. Where applicable, a protein inhibitor cocktail was added to the reaction (1x final concentration) to prevent the degradation of unmodified precursors. All reactions were incubated at 30 °C for 6 h before further analysis of the synthesized peptides and enzymes. If required, the reaction volume of CFE can be scaled up accordingly. For Western-blot analysis, His-tag labeled peptides/enzymes were visualized by using the primary antibody His-Tag Mouse Monoclonal Antibody (catalog number: 66005-1-Ig, 1:10000 dilution, Proteintech) and the secondary antibody HRP-Conjugated Affinipure Goat Anti-Mouse IgG(H + L) (catalog number: SA00001-1, 1:10000 dilution, Proteintech).

### Cell-free synthesis and purification of salivaricin B

First, modified precursor was synthesized by co-expression of SboA and SboM. To do this, two plasmids of pJL1-sboA and pET28a-sboM were added to the CFE reactions. Second, to synthesize matured peptide, namely, salivaricin B, a third plasmid (pET28a-sboT) was added to the above reaction for co-expression and cell-free expressed SboT can remove the leader peptide from SboA to form the matured peptide (salivaricin B). The total CFE reaction volume was 300 μL, which was distributed to three 2-mL tubes (i.e., 100 μL per tube) in parallel. All reactions were incubated at 30 °C with shaking (800 rpm in a thermal block shaker) for 6 h. After the reaction, three parallel CFE mixtures were combined for product identification. The modified precursor (6xHis-tagged SboA) was purified using the nickel affinity chromatography method as described above. The resulting sample was further desalted by using a C18 ZipTip and eluted with 40%, 50%, and 60% acetonitrile (ACN) containing 0.1% trifluoroacetic acid (TFA). For the matured peptide (salivaricin B), the combined reaction mixture

was centrifuged ($20,000 \times g$, 10 min) and the supernatant was desalted the same as above described. The eluted samples were then analyzed by MALDI-TOF-MS and LC-MS/MS.

To purify salivaricin B, 20 of CFE reactions were performed, each with a reaction volume of 100 μL. After the reaction (30 °C, 800 rpm, 6 h), CFE mixtures were combined (2000 μL) for salivaricin B purification. ACN was initially added to the combined mixture with a final concentration of 40% to precipitate salts and proteins. Then, the mixture was centrifuged at $20,000 \times g$ for 10 min to remove the debris. Afterward, the supernatant was further applied to reverse phase HPLC to purify salivaricin B, which is the same as described above for the preparation of standard salivaricin B.

### N-Ethylmaleimide (NEM) assay

NEM assay was performed to determine the degree of cyclization of modified lanthipeptides[20]. Samples were first treated with 3 mM tris(2-carboxyethyl) phosphine (TCEP) in HEPES buffer (50 mM, pH 7) for 20 min to ensure the complete reduction of free cysteine. Then, a freshly prepared NEM solution (50 mM in ethanol) was added with a final concentration of 3 mM. After reaction at room temperature for 1 h, the mixture was centrifuged and the pellet was resuspended in methanol for MALDI-TOF-MS analysis.

### Generation of salivaricin B variants

pJL1-derived plasmids harboring mutated *sboA* sequences were used as gene templates for synthesizing precursors and salivaricin B variants. To synthesize mutated precursors (Q6 or L20), 15 μL of CFE reactions were performed. Those mutated precursors, which can be expressed in CFE and confirmed with Western-blot and MALDI-TOF-MS, then were co-expressed with SboT (pET28a-sboT) individually to form matured salivaricin B variants. Note that here SboM-enriched cell extract was used for CFE and thus SboM was not co-expressed. The synthesis of salivaricin B variants was analyzed by MALDI-TOF-MS. To quantify salivaricin B variants, cell-free synthesized peptides were first purified as described in the section of "Cell-free synthesis and purification of salivaricin B". Briefly, ACN was added to the CFE mixture with a final concentration of 40% to precipitate salts and proteins. After centrifugation, the supernatant was applied to reverse phase HPLC to purify salivaricin B variants, which is the same as described above for the preparation of standard salivaricin B. Afterward, the purified samples were analyzed by HPLC and the yields of salivaricin B variants (Supplementary Table 5) were calculated according to a calibration curve, which was prepared by using standard salivaricin B at different concentrations (0.01, 0.05, 0.1, 0.5, and 1 mM).

### Antimicrobial activity assay

Two approaches were used for testing antimicrobial activity, including solid agar plate diffusion and liquid medium cultivation. In the diffusion assay, the overnight cultivated indicator strain *S. aureus* RN4220 was mixed with the melted LB agar medium (a starting $OD_{600}$ of 0.05). The mixture was poured onto the plate and allowed to solidify at room temperature for 30 min. Then, sterile paper disks (diameter 6 mm) were placed on the surface of the agar plate and 1 mM of each peptide was added to the paper disk. The plates were incubated at 37 °C for 16 h, and the diameter of the zone of inhibition (DZI, in mm) was measured for comparison. In the second approach, the assay was performed in 96-well plates, and cell-free synthesized mature peptides were directly used without purification. For the samples of salivaricin B variants, 80 μL of each sample (diluted if necessary) was mixed with 120 μL of LB medium, containing each variant at a roughly final concentration of 6−7 μM and the indicator strain *S. aureus* RN4220 (a starting $OD_{600}$ of 0.02). For the samples of uncharacterized (Lan 1 - Lan 10) and three known (lacticin 481, ruminococcin A, and nukacin ISK-1) lanthipeptides, 80 μL of the cell-free reaction mixture was added directly to the liquid cultivation without quantification. The 96-well

plates were incubated at 37 °C and 150 rpm for 12 h. During the cultivation, $OD_{600}$ values were measured every 2 h and $\Delta OD_{600}$ was calculated to compare the antimicrobial activity of tested samples. In all assays, kanamycin was used as a positive for comparison.

**Minimal inhibitory concentration (MIC) and IC$_{50}$ determination**
MIC values were determined by using a broth microdilution method[91]. Overnight cultures of each indicator strain were diluted to $5 \times 10^5$ CFU mL$^{-1}$ using fresh LB medium and distributed into 96-well plates. Then, the cells were treated with twofold serial dilutions of the purified peptides (i.e., salivaricin B and variants). The total volume of each cultivation was 200 μL per well, consisting of 175 μL bacteria cells and 25 μL diluted peptides. The final concentrations of each peptide were set at 0.12, 0.25, 0.5, 1, 2, 4, 8, and 16 μM. The positive growth control wells only contained bacterial cells without addition of peptides. The plates were incubated with 130 rpm at 37 °C for *B. subtilis* and *M. luteus* and 30 °C for *L. lactic*. After incubation, the lowest peptide concentration, which inhibited 90% of the bacterial growth, was recorded as the MIC. The growth curves were used to determine IC$_{50}$ values with GraphPad Prism (version 10.2.1, GraphPad Software).

**Analytical methods**
MALDI-TOF-MS analyses were performed with an Autoflex Speed MALDI-TOF-MS instrument (Bruker). 1 μL of sample was mixed with 1 μL of matrix (α-cyano-4-hydroxycinnamic acid, CHCA) and loaded onto the polished target board. After evaporation, mass weight measurement was conducted in linear positive mode (protein calibration standard: Insulin_$[M + H]^+$_avg = 5734.5200 Da, Ubiquitin_I_$[M + H]^+$_avg = 8565.7600 Da, Cytochrom_C_$[M + H]^+$_avg = 12360.9700 Da) or reflectron positive mode (peptide calibration wide range standard mono: Bradykinin (2-9)_$[M + H]^+$_mono = 904.4675 Da, ACTH_clip (1-17)_$[M + H]^+$_mono = 2093.0862 Da, ACTH_clip (1-24)_$[M + H]^+$_mono = 2932.5878 Da, ACTH_clip (7-38)_$[M + H]^+$_mono = 3657.9289 Da, ACTH (1-39)_$[M + H]^+$_mono = 4539.2666 Da). Data were analyzed using FlexAnalysis software (version 3.4, Bruker). To analyze the peptide products by LC-MS/MS, 2 μL of samples were injected into a C18 analytical column (Acclaim™ 300, 3 μM particle size, 3 mm × 150 mm). For separation, solvent B (100%ACN + 0.1%TFA) was set at 10% for 5 min, ramped up to 70% in 50 min, and increased to 100% for 2 min at a flow rate of 0.5 mL/min. The column temperature was 35 °C. The operation was carried out using ThermoFisher Q Exactive Orbitrap MS with ESI source in positive ion mode with the following parameters: resolution, 15,000; isolation width (MS/MS), 2.5 m/z; normalized collision energy (MS/MS), 35; activation q value (MS/MS), 0.25; activation time (MS/MS), 10 ms. Fragmentation was performed using collision-induced dissociation (CID) at 35% to 70%. Data were analyzed using the Qualbrowser application of Xcalibur software (version 3.0.63, ThermoFisher Scientific). All measurements were performed in biological triplicates with representative spectra displayed in relevant figures.

**Reporting summary**
Further information on research design is available in the Nature Portfolio Reporting Summary linked to this article.

## Data availability
All data supporting the findings of this study are available within the article and its Supplementary Information, or available from the corresponding author upon request. The source data underlying Figs. 2b f, 3d, 4b and Supplementary Figs. 1, 8, 12, 13, and 16, and Supplementary Table 6 are provided as a Source Data file. Source data are provided with this paper.

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

## Acknowledgements

This work was supported by the National Key Research and Development Program of China (grant no. 2023YFA0914000 to J.L.) and the National Natural Science Foundation of China (grant no. 32171427 to W.Q.L.; grant nos. 21935002 and 52322305 to S.L.). The authors also acknowledge the Cross-Disciplinary Research Fund of Shanghai Ninth People's Hospital, Shanghai Jiao Tong University School of Medicine (grant no. JYJC202219 to J.L.). M.C.J. is supported by the National Institutes of Health (grant no. 1U19AI142780-01).

## Author contributions

W.Q.L., X.J. and J.L. designed the experiments. W.Q.L. and X.J. performed the experiments. H.X. and S.H. prepared cell extracts and reagents. F.B., Y.Z. and X.Z. performed in vivo experiments. W.Q.L., Y.L., S.L. and J.L. analyzed the data and prepared the illustrations. Y.L., S.L., M.C.J. and J.L. wrote the manuscript with input from all authors. J.L. and M.C.J. contributed to the project's conception. J.L. conceived and supervised the project.

## Competing interests

The authors declare no competing interests.
