## [Peer Review File · Nature Communications]

Cell-free biosynthesis and engineering of ribosomally synthesized lanthipeptidesEditorial Note: This manuscript has been previously reviewed at another journal that is not operating a transparent peer review scheme. This document only contains reviewer comments and rebuttal letters for versions considered at Nature Communications.

Reviewers' Comments:

Reviewer #1:

Remarks to the Author:

The authors have responded strongly to all three reviewers, and addressed their comments appropriately. The revisions to the title, abstract and discussion have significantly improved the specifics of the work, which were broad before. I also particularly like the work investigating the deletion of the peptidases, and the new data showing the expression of a complex RiPP BGC. I think the paper is a strong publication highly suitable for Nature Communications, and I commend the authors rebuttal.

Reviewer #2:

Remarks to the Author:

This revised manuscript is much improved compared to the original version in terms of providing other literature examples of using CFE for RiPP production and in terms of acknowledging prior studies on demonstrating the substrate tolerance of the LanM enzymes of the lactacin 481 subgroup of lanthipeptides. In addition, with Nat Commun the intended journal is now better aligned with the degree of novelty of the study. As noted previously, the approach described in this study most certainly provides an interesting new technology that is of interest to the large RiPP community. After a few small suggested revisions shown below that will improve the presentation of the study, I recommend publication of this work in Nat Commun.

Suggested changes:

p. 5 "This observation is in agreement with the presence of the LctT protease,". LctT is not present in the experiments the authors performed. Please change to: "This observation is in agreement with previous studies on the LctT protease,"

p. 5 analine should be alanine

p. 5: "we synthesized both products by expressing their precursor peptides (each with a trypsin cleavage site) in vivo," I suggest changing in vivo to in E. coli which is more informative and makes clear the authors did not use by S. salivarius K12 which the authors talked about in the previous sentence.

"a complete three-cyclization of each peptide was also confirmed" I suggest changing this to: "the desired formation of three thioether rings was confirmed by"

p. 7, "analogs except for salivaricin B-1" change to "analogs in addition to salivaricin B-1"

When mentioning the deletion of PepN, it would be good to refer to PMID: 30679276 in which the authors show that PepN like protease removes the leader peptide of class III lanthipeptides.

Supplementary table 5. I think that the concentration yields are reported with too many significant figures. If the authors really believe the values after the period are reliable, then they should provide error bars. I suggest the authors at the very least delete values in the hundredths values and perhaps even the tenths values.

Reviewer #3:

Remarks to the Author:

Dear authors,

In this revised article, the authors have justified and satisfactorily answered the queries raised by this reviewer regarding the comparison of this CFE expression system with the commonly used in vivo co-expression strategy for different RiPP clusters.

In response to a concern raised by this reviewer, authors have mentioned that although they did not quantify each uncharacterized peptide which could be challenging, they added the same volume (80 μ L) of each sample for the antimicrobial activity assay. As the primary goal of the work was to establish a qualitative proof-of-concept CFE method for production various lanthipeptide variants, absolute quantification may not be that necessary for this study. However, some partial quantification is reported for some variants.

To address another comment, they have explained various potential strategies for enhanced peptide/protein expression in case some proteins in the selected cluster are not properly expressed and slow down the entire process. The response was satisfactory.

Most importantly, to validate the general applicability of the UniBioCat method as raised by this reviewer, the authors have performed additional experiments to successfully express another subclass of RiPPs, called glycocins (glycosylated peptides) that require posttranslational modification by glycosylation in addition to the lanthipeptide salivaricin B. This shows that method could be applied to other subclasses of RiPPs.

In addition, the comments raised on IC₅₀, and MIC determination of the modified peptides were also addressed properly by using a comprehensive approach with 3 additional bacterial strains. The discussion section was also rewritten to accommodate the comments on advantages and demerits of this method as raised by this reviewer.

Overall, the revised manuscript looks comprehensive and detailed with sufficient experimental validations for the CFE approach for producing and engineering of ribosomally synthesized lanthipeptides. The manuscript is recommended for possible publication.

Point-by-point response to reviewers' comments:

Comments in italics

Responses in blue

Reviewers' Comments:

Reviewer #1 (Remarks to the Author):

The authors have responded strongly to all three reviewers, and addressed their comments appropriately. The revisions to the title, abstract and discussion have significantly improved the specifics of the work, which were broad before. I also particularly like the work investigating the deletion of the peptidases, and the new data showing the expression of a complex RiPP BGC. I think the paper is a strong publication highly suitable for Nature Communications, and I commend the authors rebuttal.

Response: We thank you for your positive feedback and supporting publication of our manuscript.

Reviewer #2 (Remarks to the Author):

This revised manuscript is much improved compared to the original version in terms of providing other literature examples of using CFE for RiPP production and in terms of acknowledging prior studies on demonstrating the substrate tolerance of the LanM enzymes of the lacticin 481 subgroup of lanthipeptides. In addition, with Nat Commun the intended journal is now better aligned with the degree of novelty of the study. As noted previously, the approach described in this study most certainly provides an interesting new technology that is of interest to the large RiPP community. After a few small suggested revisions shown below that will improve the presentation of the study, I recommend publication of this work in Nat Commun.

Response: We thank you for your positive feedback and supporting publication of our manuscript.

Suggested changes:

p. 5 "This observation is in agreement with the presence of the LctT protease,". LctT is not present in the experiments the authors performed. Please change to: "This observation is in agreement with previous studies on the LctT protease,"

Response: Thank you. The sentence has been changed according to your suggestion.

p. 5 analine should be alanine

Response: Thank you. Corrected.

p. 5: "we synthesized both products by expressing their precursor peptides (each with a trypsin cleavage site) in vivo," I suggest changing in vivo to in E. coli which is more informative and makes clear the authors did not use by S. salivarius K12 which the authors talked about in the previous sentence.

Response: Thank you for this suggestion. Corrected.

“a complete three-cyclization of each peptide was also confirmed” I suggest changing this to: “the desired formation of three thioether rings was confirmed by”

Response: Thank you. The sentence has been changed according to your suggestion.

p. 7, “analogs except for salivaricin B-1” change to “analogs in addition to salivaricin B-1

Response: Thank you. Corrected.

When mentioning the deletion of PepN, it would be good to refer to PMID: 30679276 in which the authors show that PepN like protease removes the leader peptide of class III lanthipeptides.

Response: We have changed the reference to the suggested one (PMID: 30679276).

Supplementary table 5. I think that the concentration yields are reported with too many significant figures. If the authors really believe the values after the period are reliable, then they should provide error bars. I suggest the authors at the very least delete values in the hundredths values and perhaps even the tenths values.

Response: Thank you for this suggestion. We have updated the yields by deleting values in the tenths values.

Reviewer #3 (Remarks to the Author):

Dear authors,

In this revised article, the authors have justified and satisfactorily answered the queries raised by this reviewer regarding the comparison of this CFE expression system with the commonly used in vivo co-expression strategy for different RiPP clusters.

In response to a concern raised by this reviewer, authors have mentioned that although they did not quantify each uncharacterized peptide which could be challenging, they added the same volume (80 μ L) of each sample for the antimicrobial activity assay. As the primary goal of the work was to establish a qualitative proof-of-concept CFE method for production various lanthipeptide variants, absolute quantification may not be that necessary for this study. However, some partial quantification is reported for some variants.

To address another comment, they have explained various potential strategies for enhanced peptide/protein expression in case some proteins in the selected cluster are not properly expressed and slow down the entire process. The response was satisfactory.

Most importantly, to validate the general applicability of the UniBioCat method as raised by this reviewer, the authors have performed additional experiments to successfully express another sub-class of RiPPs, called glycosins (glycosylated peptides) that require posttranslational modification by glycosylation in addition to the lanthipeptide salivaricin B. This shows that method could be applied to other subclasses of RiPPs.

In addition, the comments raised on IC₅₀, and MIC determination of the modified peptides were also addressed properly by using a comprehensive approach with 3 additional bacterial strains. The discussion section was also rewritten to accommodate the comments on advantages and demerits of this method as raised by this reviewer.

Overall, the revised manuscript looks comprehensive and detailed with sufficient experimental validations for the CFE approach for producing and engineering of ribosomally synthesized lanthipeptides. The manuscript is recommended for possible publication.

Response: We thank you for your positive feedback and supporting publication of our manuscript.